# Motivations and Barriers to Participation in a Randomized Trial on Melanoma Genomic Risk: A Mixed-Methods Analysis

**DOI:** 10.3390/jpm12101704

**Published:** 2022-10-12

**Authors:** Gabriela Mercado, Ainsley J. Newson, David Espinoza, Anne E. Cust, Amelia K. Smit

**Affiliations:** 1The Daffodil Centre, The University of Sydney, A Joint Venture with Cancer Council NSW, Sydney 2006, Australia; 2Sydney Health Ethics, Sydney School of Public Health, Faculty of Medicine and Health, The University of Sydney, Sydney 2006, Australia; 3NHMRC Clinical Trials Centre, The Faculty of Medicine and Health, The University of Sydney, Sydney 2006, Australia; 4Melanoma Institute Australia, The University of Sydney, Sydney 2006, Australia; 5Sydney School of Public Health, Faculty of Medicine and Health, The University of Sydney, Sydney 2006, Australia

**Keywords:** genomic, polygenic trait, decision making, community participation, risk communication, clinical trial

## Abstract

The evolution of polygenic scores for use in for disease prevention and control compels the development of guidelines to optimize their effectiveness and promote equitable use. Understanding the motivations and barriers to participation in genomics research can assist in drafting these standards. We investigated these in a community-based randomized controlled trial that examined the health behavioral impact of receiving personalized melanoma genomic risk information. We examined participant responses in a baseline questionnaire and conducted interviews post-trial participation. Motivations differed in two ways: (1) by gender, with those identifying as women placing greater importance on learning about their personal risk or familial risk, and how to reduce risk; and (2) by age in relation to learning about personal risk, and fear of developing melanoma. A barrier to participation was distrust in the handling of genomic data. Our findings provide new insights into the motivations for participating in genomics research and highlight the need to better target population subgroups including younger men, which will aid in tailoring recruitment for future genomic studies.

## 1. Introduction

Integrating genomic information, such as polygenic scores, into risk assessments to inform personalized prevention and early detection is a promising strategy for reducing the burden of cancers [1]. Successful implementation of such approaches will require the development of standards and procedures to ensure equitable access, uptake, consent and communication processes [2]. Understanding the motivations and barriers among the general population to participating in research related to receiving personal polygenic risk information may inform design and recruitment strategies for future studies, including the identification of groups that may benefit from tailored approaches and support. Research on motivations and barriers to participating in genomics research have typically targeted specific population subgroups, such as those with a personal or family history of cancer [3]. These studies have shown that key motivating factors include the ability to predict personal risk, inform management and benefit families, and barriers include concerns about confidentiality, utility, psychological harm [3]. Another group studied is those undertaking whole genome sequencing with the potential to receive risk results for a wide variety of possible health conditions [4]. In primary care settings, studies have shown that individual characteristics impact the decision to accept or refuse participation in genomic research, for example, Hay et al., found that participants with higher perceived skin cancer risk or interest in learning about genes were more likely to participate in a study on skin cancer genetic testing in primary care [5]. However, limited research on providing polygenic risk information has been conducted in a community context, to examine barriers and motivations among the general population [6]. Offering polygenic testing and personal risk information in a community setting includes individuals with a broad range of characteristics (e.g., people with and without a family history of disease, wide age range, individuals who may be symptomatic or asymptomatic for other conditions), unlike research studies in healthcare settings or higher risk populations, which typically focus on patients with a strong personal or family disease history and knowledge or interest in personal disease risk. This study provides evidence towards addressing the gap in research on providing polygenic risk information to the broader community.

We aimed to examine the motivations and barriers to participating in the Melanoma Genomics *Managing Your Risk Study*, a community-based randomized controlled trial that assessed the impact of receiving (or not receiving) personal melanoma genomic risk information, on behavioral, psychosocial, ethical and economic outcomes [7].

## 2. Materials and Methods

### 2.1. Participants

The methods and design of the *Managing Your Risk Study* have been described elsewhere [7,8]. Briefly, potential participants aged 18–69 years were sampled to be representative of Australians’ State and Territory locations of residence and were sent a study invitation pack via the Australian Government’s Department of Human Services’ Medicare database. Other eligibility requirements were full or part European ancestry and no personal history of melanoma. Individuals were randomized into either intervention or control groups. Those in the intervention group received their personal melanoma risk based on a polygenic score, communicated in a personal booklet and phone call with a genetic counsellor, and presented as a remaining lifetime risk percentage and risk category (low, average, high). Participants in both intervention and control groups received an educational booklet on melanoma, but controls did not receive their personal melanoma risk.

### 2.2. Quantitative Data Collection and Analysis

Decliners to participation in the study had the option of providing a reason for this by returning a brief form to the research team via postal mail, which was included in the invitation package alongside the Participant Information Statement. On the decline form, they could provide their age, gender, postcode and an open-ended response for detailing the reason for decline.

Individuals who consented to take part in the *Managing Your Risk Study* (*N* = 1024) completed a questionnaire at baseline that contained five items on their motivations for participating in the trial [9], including: *an interest in learning more about my risk of developing melanoma; an interest in learning more about my family’s risk of developing melanoma; a fear of developing melanoma in the future, a desire to help cancer research; a desire to understand how to reduce my risk of developing melanoma*. There was no variable linked to all five motivating factors. Participants were asked to rate the importance of each item on a 3-point Likert scale (1 = not important, 2 = somewhat important, 3 = important). Questionnaire responses were analyzed by gender, age, education, and family history, which have been shown to be relevant factors for melanoma genomics research study participation [10]. These variables were collected via the baseline questionnaire using previously published measures (gender: *Are you:* response options: *male/female/other*; Age: *What is your date of birth*?; education: *What is the highest educational level you have completed?* response options: *Primary school (or equivalent), High school (or equivalent), Certificate/diploma, University degree*; family history: *Has any first-degree blood relative (parent, child, brother, sister) ever had a melanoma*? response options: *yes/no/I don’t know*) [11]. Questionnaire data were analyzed using SPSS. Differences in proportions were obtained using χ^2^ tests and a two-sided alpha of 5% was applied to interpretation of results.

### 2.3. Qualitative Data Collection and Analysis

Selected participants who completed the *Managing Your Risk Study* were invited to participate in semi-structured qualitative interviews to explore their experience in the trial, including their motivation for participating. Interviewed participants (*N* = 40) were purposively recruited to include a range of characteristics by age, sex, genomic risk level (for intervention arm participants) and geographical location. The interviews were conducted by a trained researcher and were audio-recorded and transcribed. For this analysis, two researchers (AKS, GM) undertook thematic analysis of the interview transcript data, guided by a coding framework to develop themes related to motivations or barriers to participation in the trial [12]. Participants typically raised motivations and barriers in response to facilitator questions about participation in the trial, specifically: “*Recently you participated in a study about managing your risk of melanoma. As part of the study, you received information about your chances of developing melanoma based on your genetic risk make-up. Can you tell me a bit about your experience of the study*?” Any discrepancies in coding were discussed with the research team. The coding was facilitated by NVivo qualitative data analysis software (QSR International Pty Ltd. Version 11).

For this analysis, we report and interpret the findings from the qualitative and quantitative analysis together [12] (i.e., data triangulation), which is a recommended methodology for exploring the context of participant experiences [13].

Ethics approval was obtained from the Human Research Ethics Committee at The University of Sydney and participants gave written informed consent.

## 3. Results

A total of 1024 participants gave consent and completed the baseline questionnaire and 40 took part in a qualitative interview (*N* = 40; Table 1). During recruitment, 87 individuals actively declined participation and most provided a reason (summarized in Table 2). Of the participants who declined, 48 declined due to being ineligible, 62 individuals provided their age (mean = 51 years, standard deviation = 16.6) and 80 provided their gender (*N* = 29 female).

### 3.1. Motivations and Perceived Benefit for Participation

In the baseline questionnaire, overall, the factor that was the most important in motivating participation was “a desire to help cancer research” (rated as not important, *N* = 45; somewhat important, *N* = 282; important, *N* = 697; Figure 1D) and the least important was “a fear of developing melanoma in the future” (rated as not important, *N* = 290; somewhat important, *N* = 468; important, *N* = 269; Figure 1C). We identified differences in responses to questions on the factors influencing study participation between men and women. People identifying as women were more likely to rate as ‘important’ items related to learning about their personal risk (women 74%, men 62%; *p* ≤ 0.001, Figure 1A) or familial risk (70%, 58%; *p* ≤ 0.001, Figure 1B), and how to reduce risk (women 67%, men 55%; *p* ≤ 0.001, Figure 1E). In the semi-structured interviews, women described an increase in their awareness of risky sun-related behaviors and an intention to modify them after their experience in the study.

Female (average risk, age 67) “… (participating) made me more aware, even though I’m much older, more aware for the next generations of where this is all going to and how to protect them…”.

Personal skin cancer risk factors, such as hair color and time spent outdoors, were other motivating factors for participation in the trial that were identified in the interviews:

Female (control group, age 45) “Well I’m always really keen to help with anything that furthers our knowledge of these sort of things. I suppose I have a personal interest in terms of my skin type and my situation meaning that I’m out in the sun a lot. I suppose one, for my own personal information, but two, just a greater information, the more we can know about these things, the more we can better manage them and prevent them. So yeah, I was happy to be involved”.

Female (high risk, age 42) “I was happy to be part of the study because I’m a fair-haired, ginger, English person so I know how important it is for these things to go ahead”.

Compared to participants aged 45–69 years, younger participants (aged 18–44 years) were more likely to rate ‘learning about their personal risk’ (76% vs. 61%; *p* ≤ 0.001, Figure 1A), ‘fear of developing melanoma’ (36% vs. 23%; *p* ≤ 0.001, Figure 1C) as important. We observed similar patterns when we further stratified the data by age and sex. ‘Fear of developing melanoma’ was less of a motivating factor for people with a university degree compared to those with school-only educational attainment (rated as important by 28% and 33% respectively; *p* ≤ 0.001). Among all questionnaire items, ‘a desire to help cancer research’ was the strongest motivator for participation across all groups with 88% of participants overall rating it as ‘important’ (Figure 1D). In the qualitative interviews, participants linked helping cancer research with the potential to also benefit themselves in the future:

Male (low risk, age 43) “my view is that if you want to benefit from the treatment and the advances, if you get asked to participate in something like this, you probably should do it”.

Participants did not express enduring fears regarding a future melanoma diagnosis in the interviews. Some did discuss that knowing a friend or family member who had been diagnosed with melanoma was a motivating factor:

Female (low risk, age 67) “I had a friend who died of melanoma, so the study itself was very useful for me to run through the questions, (it) made me think about stuff (and) I found it very useful”.

### 3.2. Barriers to Participation

Reasons for declining to participate in the research trial are listed in Table 2. Among those who actively declined participation and provided a reason for this (N = 83), more than half of the reasons listed were related to the trial eligibility criteria. Other reasons included nature of employment, pre-existing health conditions, or English language fluency. Some participants reported a distrust in genetic data collection and storage and a fear that insurers would access the information and increase risk-rated insurance premiums. The only barrier to participation that was identified in the semi-structured interviews was concern about insurance:

Male (high risk, age 50)—“I think the only concerning thing was maybe at the start of the study they said that it might increase (risk-rated) insurance premiums because you have to declare it”.

Male (high risk, age 49)—“My only reservation in doing it was that I wanted to be able to tick the box on my insurance form that says I’ve never had a genetic test and I don’t know what the results are and therefore you can’t discriminate against me when you’re providing insurance as I get older and so I was very keen to understand how the security was being handled and I was very much satisfied by that. So I was really keen to participate and the information was good”.

## 4. Discussion

This research is the first to look at motivations for, and barriers to, participating in a personalized melanoma risk study in a community-based setting. Past studies that have explored motivations and barriers found that altruism and learning about one’s personal risk were common reasons for participating in genomics research [4,14,15]. Although our study focused on a melanoma polygenic score rather than highly penetrant mutations, we found that ‘desire to help cancer research’ remained a strong motivator among participants.

Other motivators that we observed varied by demographic characteristics. The desire women had to learn about personal risk and the intention to use it to improve their health is consistent with other skin cancer-focused research studies that have shown women are more likely than men to adopt behavior changes to reduce the likelihood of developing melanoma [16,17,18]. Familial benefit as a motivator among female participants has been previously cited by Goodman et al., and could be explained by the role women have ascribed to them in families as the ‘gatekeeper’ of their children’s health [14]. The differences in motivations by gender identified in our analysis may also relate to the lower participation rates previously reported in the Managing Your Risk study (18–44-year-old men: 1.4% and women: 3.0%; 45–69 year old men: 4.5% and women: 6.7%) [19]. With relation to age, a high awareness of skin cancer susceptibility among the younger Australian population has been noted by other studies [20,21], and supports younger participants rating ‘learning about personal risk’ and ‘fear of developing melanoma’ as important motivators for joining the study. Providing information on the potential to learn about personal disease risk from genomic research studies during recruitment may be a strategy for maximizing participation from younger populations. Other variables that were not part of this analysis but have been linked to participation in other research studies include information seeking style [22] and having children [15]. These may also have influenced motivations to participate in this trial.

For those who declined participation in the study, it was important for us to understand the barriers that were impeding their participation. Distrust in genetic data collection and fear of genetic discrimination as a barrier to participation have been noted in previous genetic and genomic research studies [6,23]. In Australia, at the time of writing there is a partial moratorium (ban) limiting the use of genetic test results in life insurance underwriting. This moratorium is industry self-regulated and applies only to policies below certain financial limits (e.g., $500,000 of death cover) [24]. Some decreases in patients delaying/declining testing after the moratorium’s introduction has been reported [24]. Currently, there are no specific insurance-related guidelines about the use of polygenic scores. Our findings support the need for guidelines that permit individuals to not disclose genetic results for risk-rated insurance to assist in reducing this barrier [25]. Future genomics research could also address this during recruitment by employing strategies such as providing transparent information about who will benefit from genomic data access, the option for participants to withdraw data from research studies and information about who is using the data, and for what purpose [26].

Another consideration in our study was that only people with European ancestry were eligible to participate, which is due largely to interpretation databases being composed of data drawn from people with white ethnicity. While this barrier was inherent to the study itself (and is thus not one reported or experienced by our participants), it highlights an important and pressing need to improve the representation of diverse ethnic groups in genomic databases [27]. This inherent barrier also prevented information regarding participation in genomic research being obtained from other ethnic groups. Existing research on barriers to research participation (from both genomic and non-genomic research studies) could be drawn on to inform how polygenic score studies can be optimized for accessibility to participants from diverse ethnic, and socio-economic backgrounds [28,29,30], for example, through the tailoring of study invitations and intervention materials according to diverse cultural and language contexts [31].

A limitation of our study is that we have only been able to explore motivations and barriers to participation among those who participated in the study. Only active decliners provided a reason for not participating, and over half of these were due to being ineligible. Therefore, there may be other potential barriers beyond concerns about insurance, which was the only barrier identified in the semi-structured interviews. Further, although the probing question used in the interviews prompted the participants to discuss motivations and barriers to taking part in the trial, the wording of the question did not explicitly ask participants about these factors. In addition to further exploring barriers to participation in genomic research among groups that decline to take part, the framing of probing questions in future research could focus more specifically on barriers and motivations. A strength of our study was its discernment of views and attitudes from over 1000 participants from all States and Territories across Australia, which was balanced for gender and age. Studies that focus on providing personalized genomic risk information should continue to assess and actively work to counter the barriers and facilitators faced by different groups.

In conclusion, the results presented in this paper can assist genomics researchers in developing personalized strategies for recruiting trial participants, especially groups that tend to have lower participation rates such as younger men. Understanding motivators and barriers that different groups face when deciding whether to participate in genomic research will aid in designing studies that are accessible to various individuals not just a one-size fits all program.

## Figures and Tables

**Figure 1 jpm-12-01704-f001:**
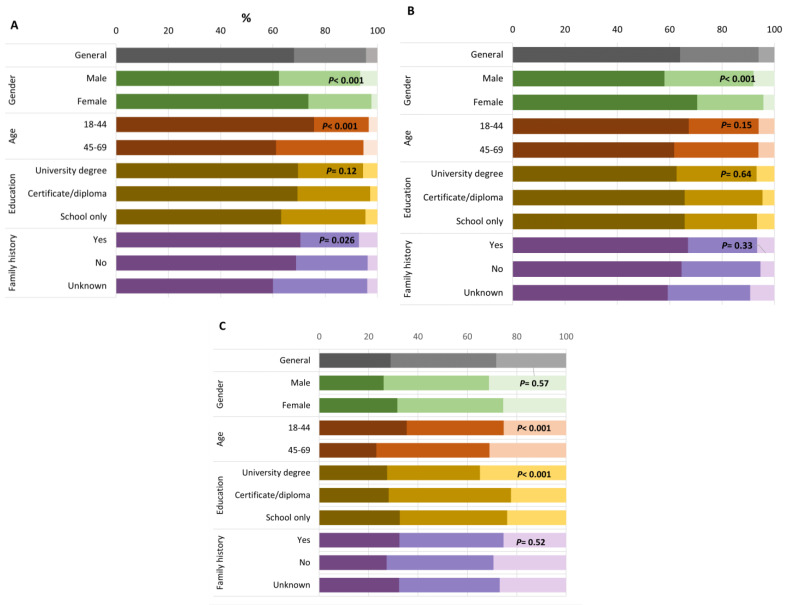
Importance of factors in deciding to participate in the *Managing Your Risk Study*. (**A**) An interest in learning more about my risk of developing melanoma; (**B**) An interest in learning more about my family’s risk of developing melanoma; (**C**) A fear of developing melanoma in the future; (**D**) A desire to help cancer research; (**E**) A desire to understand how to reduce my risk of developing melanoma. Darkest color = important, medium color = somewhat important, lightest color = not important. *p*-values compare proportions of importance ratings across groups.

**Table 1 jpm-12-01704-t001:** Participant demographic characteristics.

Characteristics	Participants Who Completed the Baseline Questionnaire (*N* = 1024 ^1^)	Interviewed Participants (*N* = 40 ^2^)
	*N* (%)	*N* (%)
**Gender**		
Female	522 (51.0%)	24 (60.0%)
Male	502 (49.0%)	16 (40.0%)
**Age group**		
18-44 years	483 (47.2%)	15 (37.5%)
45-69 years	541 (52.8%)	25 (62.5%)
**Education**		
University degree	458 (44.7%)	20 (50.0%)
Certificate/diploma	327 (31.9%)	11 (27.5%)
School level (or equivalent)	239 (23.4%)	9 (22.5%)
**Family history of melanoma**		
Yes	197 (19.2%)	7 (17.5%)
No	697 (68.1%)	28 (70.0%)
Unknown	130 (12.7%)	5 (12.5%)
**State of residence**		
NSW	289 (28.2%)	10 (25.0%)
QLD	210 (20.5%)	6 (15.0%)
WA	106 (10.4%)	7 (17.5%)
NT	6 (0.6%)	0
TAS	45 (4.4%)	2 (5.0%)
VIC	293 (28.6%)	12 (30.0%)
SA	56 (5.5%)	3 (7.5%)
ACT	19 (1.9%)	0
**Genomic risk category (intervention arm only)**	*N* = 509	*N* = 20
Lower than average	107 (21.0%)	10 (50%)
Average	267 (52.5%)	3 (15%)
Higher than average	135 (26.5%)	7 (35%)

^1^ Data is missing for one person who withdrew consent. ^2^ The interviews were conducted after the parent study was completed.

**Table 2 jpm-12-01704-t002:** Reasons for actively declining participation in the study.

Reason	*N* (%) (*N* = 87)	Description	Example Quote
**Eligibility**			
Diagnosed with melanoma	19 (22)	Participants had a prior history of melanoma and/or had a melanoma excised	“I would like to take part in this study but am unsuitable as I have had a melanoma.”
Overseas or a different state	21 (24)	Participants are currently living abroad or in a different state	“Received letter advising that addressee now lives overseas and is unable to participate.”
Age	3 (3)	Participants are above the age cut off for the research	“Called to notify that he had just recently turned 70 years old, so now not eligible.”
No European ancestry	5 (6)	Participants disclosed they have no European ancestry	“I cannot take part in this study because I do not have any European ancestry.”
**Other Reasons**			
Insurance concerns	5 (6)	Participants do not want to participate because they are worried their genetic risk will be used by insurance companies	“I have declined due to having to report genetic results to insurance bodies. This would not only impact me, but also my children.”
Time constraints	5 (6)	Participants express that they are too busy to dedicate time to the research	“Sorry but I run a business and don’t have time for this research."
Distrust in genetic data collection/handling	3 (3)	Participants are worried their genetic information wouldn’t be protected/safe	“I’m simply not keen on having my genetic material collected and stored. Despite best intentions, accidents and breaches can still occur.”
Other health conditions	3 (3)	Participants disclosed that they are currently undergoing other health problems and cannot take on other commitments	“I do not wish to do this due to me suffering renal carcinoma.”
Disability	4 (5)	Participant and/or carer informed that they have a disability that prevents them from participating	“Our son has an intellectual and partly physical disability and cannot participate in your study.”
No interest	2 (2)	Participant disclosed no interest in taking part in the research	“Not interested in taking part in study.”
Other	8 (9)	Reason disclosed does not fall under previous categories	“I am not in a position to pay any fees.”
			“English not good.”
			“Declining due to not being able to wear that wristband to work due to being a mechanic.”
Declined but no reason given	4 (5)		

## Data Availability

De-identified data presented in this study are available on request from the corresponding author and sharing will be subject to ethics approval. The data are not publicly available due to ethics requirements.

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
