# Peer review of "Motivations and Barriers to Participation in a Randomized Trial on Melanoma Genomic Risk: A Mixed-Methods Analysis"

_jpm, 2022, doi:10.3390/jpm12101704_

Round 1

Reviewer 1 Report

This is an interesting paper, reporting on the reasons or not for participation in the Managing your Risk study. The findings are informative for understanding about involvement in these types of studies. There are few minor points for clarification.

1. The focus of the study findings is unclear. i.e. is this about PGS related research as stated in the discussions or genomic research in general as stated in the introductions. My understanding is that it is about the later, but maybe this can be clarified. 

2. It would also be useful in the introduction to explain a little bit how community settings differ from other research settings described. I assume this is due to the wider age range and non-specific health care context. This may also help better highlight and differences and similarities between existing research that has been carried out on participation in genomic research. 

3. The authors note that the primary reason for declining to participate was not meeting study eligibility criteria. These reasons are outside of the control of individuals, hence are not directly applicable to understanding lack/unwillingness to participate. It is unclear of those who declined to participate what proportion were eligible to. Analysis should be restricted to these individuals. 

4. In relation to the above, in the discussion the authors state: "A limitation of our study was that only people with European ancestry were eligible to participate, which highlights an important and pressing need to improve the representation of ethnic groups in genomic databases". This statement makes no sense. What is the relationship between eligibility for participation and the need for databases. The limitation is that as those not of European ancestry were unable to participate in this study, it can't provide any information across ethnic groups in any factors that influence participation in genomic research. 

5. It would be interesting to reflect in the discussions whether any of these findings are specific to genomics research or are similar across other areas.

Author Response

  1. The focus of the study findings is unclear. i.e. is this about PGS related research as stated in the discussions or genomic research in general as stated in the introductions. My understanding is that it is about the later, but maybe this can be clarified.

Author response: We have revised the wording in the Introduction to address this point and to clarify that these study findings are about personal polygenic (genomic) sequencing, pages 3-4:

“Understanding the motivations and barriers among the general population to participating in research related to receiving personal polygenic risk information…

However, limited research on communicating polygenic risk information has been conducted in a community context…

Offering polygenic testing and personal risk information in a community setting…”

  1. It would also be useful in the introduction to explain a little bit how community settings differ from other research settings described. I assume this is due to the wider age range and non-specific health care context. This may also help better highlight and differences and similarities between existing research that has been carried out on participation in genomic research.

Author response: We have now addressed this point in the Introduction, pages 3-4:

“Offering polygenic testing and personal risk information in a community setting includes individuals with a broad range of characteristics (e.g. people with and without a family history of disease, wide age range, individuals who may be symptomatic or asymptomatic for other conditions), unlike research studies in healthcare settings or higher risk populations, which typically focus on patients with a strong personal or family disease history and knowledge or interest in personal disease risk. This study provides evidence towards addressing the gap in research on providing polygenic risk information to the broader community.”

  1. The authors note that the primary reason for declining to participate was not meeting study eligibility criteria. These reasons are outside of the control of individuals, hence are not directly applicable to understanding lack/unwillingness to participate. It is unclear of those who declined to participate what proportion were eligible to. Analysis should be restricted to these individuals.

Author response: We have now added more detail to the Results, page 6: “Of the participants who declined participation, 48 declined due to being ineligible”, and the Discussion, page 12: “A limitation of our study is that we have only been able to explore motivations and barriers to participation among those who participated in the study. Only active decliners provided a reason for not participating, and over half of these were due to being ineligible. Therefore, there may be other potential barriers beyond concerns about insurance, which was the only barrier identified in the semi-structured interviews.”

We note that the analysis presented in this paper does not include individuals who declined to participate due to being ineligible.

  1. In relation to the above, in the discussion the authors state: "A limitation of our study was that only people with European ancestry were eligible to participate, which highlights an important and pressing need to improve the representation of ethnic groups in genomic databases". This statement makes no sense. What is the relationship between eligibility for participation and the need for databases. The limitation is that as those not of European ancestry were unable to participate in this study, it can't provide any information across ethnic groups in any factors that influence participation in genomic research.

Author response: The relationship between eligibility for this study and ‘the need for [diverse] databases’ is that databases must be representative of the diversity of populations in order to facilitate a reliable polygenic score. At the time of recruitment for this study, databases did not meet this requirement. We have amended the sentence to clarify our meaning, page 11: “A limitation of our study was that only people with European ancestry were eligible to participate. This barrier is due largely to interpretation databases being composed of data drawn from people with white ethnicity. While this barrier was inherent to the study itself (and is thus not one reported or experienced by our participants), it highlights an important and pressing need to improve the representation of diverse ethnic groups in genomic databases. This inherent barrier also prevented information regarding participation in genomic research being obtained from other ethnic groups.”

  1. It would be interesting to reflect in the discussions whether any of these findings are specific to genomics research or are similar across other areas.

Author response: In the Discussion, we have added further detail on the other research studies that we reference to highlight similarities with other research areas (mainly skin cancer and melanoma related research) and the genomics context:

Page 10: “The desire women had to learn about personal risk and the intention to use it to improve their health is consistent with other skin cancer-focused research studies…

Family benefit as a motivator among female participants in genetic research studies has been previously cited by Goodman et al…

“Distrust in genetic data collection and fear of genetic discrimination as a barrier to participation have been noted in previous genetic and genomic research studies…”

Page 11: “Existing research on barriers to research participation (from both genomic and non-genomic research studies) could be drawn on…”

Reviewer 2 Report

The information in the introduction should provide more information about the motivations and barriers reportted in the former studies. 

In the methods section:

- Authors should provide the scale they used to measure the variables age, gender and education. Also, more detailed should be provided on how the family history was collected. Also, a reasoning for choosing these specific variables should be provided. 

- It would be important to add the probing question used in the qualitative interviews to solicit input about the motivation for participating.

- Atuhors should explain how they approched the integration of quantitative and qualitative methods. There are many approach in the conduct of a mixed-methods design. It would be important to infor readers about what has been done. 

In the results section: 

- Excellent idea to present all the reasons for not participating with this level details. 

- The contrast of text in Figure 1 should be improved. 

- Authors should present first in their text the general proportions of response to each factors in deciding to participate stating which factor was the most and which one was the least important in motivating the participation. 

- An important result that is not recognized in the manuscript is that there is no variable that is linked to all five motivating factors. 

- The passage "Participants did not express enduring fears regarding a future melanoma diagnosis in the interviews" is the only link between the questionnaire and the interviews. If this is possible, it would be great that authors expand on the themes that did or that did not come up related to the other of the main factors (i.e. learning about own's risk, about family's risk, desire to help research and desire to understand own's risk). 

- The presentation of the thematic analysis should be enhanced as this is probably a rich collection of different factors. 

In the discussion: 

- Authors should discuss about the other explanatory variables that were not part of their analysis that could be explaining some of the motivating factors. Risk perception, information seeking style, having children have been shown to be linked to participation. 

- Authors should recognized that there is an inherent selection bias in questioning motivations in people who did chose to participate to their study. Not something they can remediate, but certainly something worth discussing.  

- I'm not sure I am convinced by the statement that their population is one from the general population.  Could the authors bring more elements to support this aspect in their introduction and in their discussion. 

- In the discussion, few of the barriers are discussed and there is almost no recommandation for researchers on how to address these barriers or to take advantage of the most frequent motivating factors. For example, authors should expand on how they would advise researchers to adress the barrier of distrust in genetic collection and handling or time contraints in their study invitation and design. 

Author Response

Reviewer #2:

  1. The information in the introduction should provide more information about the motivations and barriers reported in the former studies.

Author response: We have added more information to the Introduction to address this point, pages 3-4:

“These studies have shown that key motivating factors include the ability to predict personal risk, inform management and benefit families, and barriers include concerns about confidentiality, utility, psychological harm (3)…

In primary care settings, studies have shown that individual characteristics impact the decision to accept or refuse participation in genomic research, for example, Hay et al found that participants with higher perceived skin cancer risk or interest in learning about genes were more likely to participate in a study on skin cancer genetic testing in primary care (5).”

In the methods section:

  1. Authors should provide the scale they used to measure the variables age, gender and education. Also, more detailed should be provided on how the family history was collected. Also, a reasoning for choosing these specific variables should be provided.

Author response: We have now incorporated these additional details as requested, page 5:

“Questionnaire responses were analyzed by gender, age, education, and family history, which have been shown to be relevant factors for melanoma genomics research study participation (10). These variables were collected via the baseline questionnaire using previously published measures (gender: Are you: response options: male/female/other; Age: What is your date of birth?; education: What is the highest educational level you have completed? response options: Primary school (or equivalent), High school (or equivalent), Certificate/diploma, University degree; family history: Has any first-degree blood relative (parent, child, brother, sister) ever had a melanoma? response options: yes/no/I don’t know) (11).”

  1. It would be important to add the probing question used in the qualitative interviews to solicit input about the motivation for participating.

Author response: We have now added the relevant question that prompted discussion of barriers and facilitators to participation in the trial during the interviews, page 6:

“Participants typically raised motivations and barriers in response to facilitator questions about participation in the trial, specifically: “Recently you participated in a study about managing your risk of melanoma. As part of the study, you received information about your chances of developing melanoma based on your genetic risk make-up. Can you tell me a bit about your experience of the study?”

  1. Authors should explain how they approached the integration of quantitative and qualitative methods. There are many approaches in the conduct of a mixed-methods design. It would be important to inform readers about what has been done.

Author response: We have now detailed the integration of the mixed-methods for this analysis, page 10:

“For this analysis, we report and interpret the findings from the qualitative and quantitative analysis together [13] (i.e. data triangulation), which is a recommended methodology for exploring the context of participant experiences [14].”

In the results section:

  1. Excellent idea to present all the reasons for not participating with this level details. The contrast of text in Figure 1 should be improved.

Author response: We have now improved the contrast of the text in Figure 1.

  1. Authors should present first in their text the general proportions of response to each factor in deciding to participate stating which factor was the most and which one was the least important in motivating the participation.

Author response: We have now added these results to the text, page 7:

“In the baseline questionnaire, overall, the factor that was the most important in motivating participation was “a desire to help cancer research” (rated as not important, N=45; somewhat important, N=282; important, N=697; Figure 1D) and the least important was “a fear of developing melanoma in the future” (rated as not important, N=290; somewhat important, N=468; important, N=269; Figure 1C).”

  1. An important result that is not recognized in the manuscript is that there is no variable that is linked to all five motivating factors.

Author response: We have now added this point to the manuscript, page 5:

“There was no variable linked to all five motivating factors.”

  1. The passage "Participants did not express enduring fears regarding a future melanoma diagnosis in the interviews" is the only link between the questionnaire and the interviews. If this is possible, it would be great that authors expand on the themes that did or that did not come up related to the other of the main factors (i.e. learning about own's risk, about family's risk, desire to help research and desire to understand own's risk). The presentation of the thematic analysis should be enhanced as this is probably a rich collection of different factors.

Author response: We have now expanded on the results from the thematic analysis as suggested by this Reviewer.

Page 7:

“Personal skin cancer risk factors, such as hair color and time spent outdoors, were other motivating factors for participation in the trial that were identified in the interviews:

Female (control group, age 45) “Well I'm always really keen to help with anything that furthers our knowledge of these sort of things. I suppose I have a personal interest in terms of my skin type and my situation meaning that I'm out in the sun a lot. I suppose one, for my own personal information, but two, just a greater information, the more we can know about these things, the more we can better manage them and prevent them. So yeah, I was happy to be involved.”

Page 8:

 “Female (high risk, age 42) “I was happy to be part of the study because I'm a fair-haired, ginger, English person so I know how important it is for these things to go ahead”

In the qualitative interviews, participants linked helping cancer research along with the potential to also benefit themselves in the future:

Male (low risk, age 43) “my view is that if you want to benefit from the treatment and the advances, if you get asked to participate in something like this, you probably should do it.”

Participants did not express enduring fears regarding a future melanoma diagnosis in the interviews. Some did discuss that knowing a friend or family member who had been diagnosed with melanoma was a motivating factor:

Female (low risk, age 67) “I had a friend who died of melanoma, so the study itself was very useful for me to run through the questions, [it] made me think about stuff [and] I found it very useful”

Page 9:

“The only barrier to participation that was identified in the semi-structured interviews was concern about insurance:

Male (high risk, age 50) – “I think the only concerning thing was maybe at the start of the study they said that it might increase [risk-rated] insurance premiums because you have to declare it.”

Male (high risk, age 49) – “My only reservation in doing it was that I wanted to be able to tick the box on my insurance form that says I've never had a genetic test and I don't know what the results are and therefore you can't discriminate against me when you're providing insurance as I get older and so I was very keen to understand how the security was being handled and I was very much satisfied by that. So I was really keen to participate and the information was good.”

In the discussion:

  1. Authors should discuss about the other explanatory variables that were not part of their analysis that could be explaining some of the motivating factors. Risk perception, information seeking style, having children have been shown to be linked to participation.

Author response: We have now addressed this in the Discussion, page 12:

“Other variables that were not part of this analysis but have been linked to participation in other research studies include information seeking style [32] and having children [16].”

  1. Authors should recognize that there is an inherent selection bias in questioning motivations in people who did chose to participate to their study. Not something they can remediate, but certainly something worth discussing.

Author response: We have now addressed this point in the Discussion, page 12:

“A limitation of our study is that we have only been able to explore motivations and barriers to participation among those who participated in the study. Only active decliners provided a reason for not participating, and over half of these were due to being ineligible. Therefore, there may be other potential barriers beyond concerns about insurance, which was the only barrier identified in the semi-structured interviews.”

  1. I'm not sure I am convinced by the statement that their population is one from the general population. Could the authors bring more elements to support this aspect in their introduction and in their discussion.

Author response: We have now added more detail about the population in this study:

Page 4,

“Briefly, potential participants aged 18-69 years were sampled to be representative of Australians’ State and Territory locations of residence and were sent a study invitation pack via the Australian Government’s Department of Human Services’ Medicare database.”

Page 11,

“A strength of our study was its discernment of views and attitudes from over 1,000 participants from all States and Territories across Australia, which was balanced for gender and age.”

  1. In the discussion, few of the barriers are discussed and there is almost no recommendation for researchers on how to address these barriers or to take advantage of the most frequent motivating factors. For example, authors should expand on how they would advise researchers to address the barrier of distrust in genetic collection and handling or time constraints in their study invitation and design.

Author response: We have now discussed these considerations in the text:

Page 10,

“Providing information on the potential to learn about personal disease risk from genomic research studies during recruitment may be a strategy for maximizing participation from younger populations.”

Page 11,

“Future genomics research could address the barrier of distrust in testing or handling of genetic information during recruitment by employing strategies such as providing transparent information about who will benefit from genomic data access, the option for participants to withdraw data from research studies and information about who is using the data, and for what purpose [26]…

Other issues such as low participation by men highlights the need to better understand the factors that motivate or pose a barrier to participation to genomics research in population subgroup

Round 2

Reviewer 2 Report

Congratulations for this new version of your manuscript. 

Knowing the exact wording of the qualitative interviews probing question, I believe it would be important for authors to report in their discussion of limitations that their question was not framed specifically to collect input on motivations and barriers to participate in genomic research. 

The authors claimed that the contrast of Figure 1 was improved, but on my end it is even worst as I'm now hardly able to read the text. 

The present format of the discussion would benefit from a thorough editing to improve its readability (i.e. the flow of ideas is not always easy to follow, there are repetitions that could be cut).

Author Response

  1. Knowing the exact wording of the qualitative interviews probing question, I believe it would be important for authors to report in their discussion of limitations that their question was not framed specifically to collect input on motivations and barriers to participate in genomic research.
    Author response: We have now addressed this point in the discussion, page 11-12:
    “Further, although the probing question used in the interviews prompted the participants to discuss motivations and barriers to taking part in the trial, the wording
    of the question did not explicitly ask participants about these factors. In addition to further exploring barriers to participation in genomic research among groups that decline to take part, the framing of probing questions in future research could focus more specifically on barriers and motivations.”

    2. The authors claimed that the contrast of Figure 1 was improved, but on my end it is even worst as I'm now hardly able to read the text.

    Author response: We have saved the Figure in higher quality files in a different format and note that Figure 1 is now shown on two pages. This is saved in the document and uploaded as individual files to the platform.

3. The present format of the discussion would benefit from a thorough editing to improve its readability (i.e. the flow of ideas is not always easy to follow, there are repetitions that could be cut)"
Author response: We have now addressed this point by revising the structure and flow of the entire discussion, and trimmed some text, pages 9-12:
“This research is the first to look at motivations for, and barriers to, participating in a
personalized melanoma risk study in a community-based setting. Past studies that have explored motivations and barriers found that altruism and learning about one’s personal risk were common reasons for participating in genomics research [4,15,16].
Although our study focused on a melanoma polygenic score rather than highly penetrant mutations, we found that ‘desire to help cancer research’ remained a strong motivator among participants.
Other motivators that we observed varied by demographic characteristics. The desire women had to learn about personal risk and the intention to use it to improve their health is consistent with other skin cancer-focused research studies that have shown women are more likely than men to adopt behavior changes to reduce the likelihood of developing melanoma [17-19]. Familial benefit as a motivator among
female participants has been previously cited by Goodman et al., and could be explained by the role women have ascribed to them in families as the ‘gatekeeper’ of their children’s health [15]. The differences in motivations by gender identified in our
analysis may also relate to the lower participation rates previously reported in the Managing Your Risk study (18–44-year-old men: 1.4% and women: 3.0%; 45-69 year old men: 4.5% and women: 6.7%) [20]. With relation to age, a high awareness of skin cancer susceptibility among the younger Australian population has been noted by other studies [21,22], and supports younger participants rating ‘learning about personal risk’ and ‘fear of developing melanoma’ as important motivators for joining the study. Providing information on the potential to learn about personal disease risk from genomic research studies during recruitment may be a strategy for maximizing participation from younger populations. Other variables that were not part of this analysis but have been linked to participation in other research studies include information seeking style [23] and having children [16]. These may also have influenced motivations to participate in this trial.
For those who declined participation in the study, it was important for us to understand the barriers that were impeding their participation. Distrust in genetic data collection and fear of genetic discrimination as a barrier to participation have
been noted in previous genetic and genomic research studies [6,24]. In Australia, at the time of writing there is a partial moratorium (ban) limiting the use of genetic test results in life insurance underwriting. This moratorium is industry self-regulated and applies only to policies below certain financial limits (e.g., $500,000 of death cover) [25]. Some decreases in patients delaying/declining testing after the moratorium’s introduction has been reported [25]. Currently, there are no specific insurance-
related guidelines about the use of polygenic scores. Our findings support the need for guidelines that permit individuals to not disclose genetic results for risk-rated insurance to assist in reducing this barrier [26]. Future genomics research could also address this during recruitment by employing strategies such as providing transparent information about who will benefit from genomic data access, the option
for participants to withdraw data from research studies and information about who is using the data, and for what purpose [27].

Another consideration in our study was that only people with European ancestry were eligible to participate, which is due largely to interpretation databases being composed of data drawn from people with white ethnicity. While this barrier was
inherent to the study itself (and is thus not one reported or experienced by our participants), it highlights an important and pressing need to improve the representation of diverse ethnic groups in genomic databases [28]. This inherent
barrier also prevented information regarding participation in genomic research being obtained from other ethnic groups. Existing research on barriers to research participation (from both genomic and non-genomic research studies) could be drawn on to inform how polygenic score studies can be optimized for accessibility to
participants from diverse ethnic, and socio-economic backgrounds [29-31], for example, through the tailoring of study invitations and intervention materials according to diverse cultural and language contexts [32].

A limitation of our study is that we have only been able to explore motivations and barriers to participation among those who participated in the study. Only active decliners provided a reason for not participating, and over half of these were due to being ineligible. Therefore, there may be other potential barriers beyond concerns about insurance, which was the only barrier identified in the semi-structured interviews. Further, although the probing question used in the interviews prompted
the participants to discuss motivations and barriers to taking part in the trial, the wording of the question did not explicitly ask participants about these factors. In addition to further exploring barriers to participation in genomic research among groups that decline to take part, the framing of probing questions in future research could focus more specifically on barriers and motivations. A strength of our study was its discernment of views and attitudes from over 1,000 participants from all States and Territories across Australia, which was balanced for gender and age. Studies that focus on providing personalized genomic risk information should
continue to assess and actively work to counter the barriers and facilitators faced by different groups.

In conclusion, the results presented in this paper can assist genomics researchers in developing personalized strategies for recruiting trial participants, especially groups that tend to have lower participation rates such as younger men. Understanding motivators and barriers that different groups face when deciding whether to
participate in genomic research will aid in designing studies that are accessible to various individuals not just a one-size fits all p
rogram.”
